# A No-go Theorem for Robust Acceleration in the Hyperbolic Plane

Linus Hamilton*        Ankur Moitra†

## Abstract

In recent years there has been significant effort to adapt the key tools and ideas in convex optimization to the Riemannian setting. One key challenge has remained: Is there a Nesterov-like accelerated gradient method for geodesically convex functions on a Riemannian manifold? Recent work has given partial answers and the hope was that this ought to be possible. Here we prove that in a noisy setting, there is no analogue of accelerated gradient descent for geodesically convex functions on the hyperbolic plane. Our results apply even when the noise is exponentially small. The key intuition behind our proof is short and simple: *In negatively curved spaces, the volume of a ball grows so fast that information about the past gradients is not useful in the future.*

## 1  Introduction

Convex optimization [24] undergirds much of machine learning, theoretical computer science, operations research and statistics. Geodesically convex optimization is a natural generalization that replaces Euclidean space with a Riemannian manifold and we require that the function we want to minimize is convex along geodesics [31, 1, 5]. It turns out that many optimization problems of interest, while non-convex in the Euclidean view, become geodesically convex when equipped with the right geometry. Some notable examples: The fastest known algorithms for computing Brascamp-Lieb constants [17, 3], and solving related problems like the null cone problem [8, 6, 7], exploit geodesic convexity. In machine learning, it arises in matrix completion [9, 28, 33], dictionary learning [11, 26], robust subspace recovery [39], mixture models [18] and optimization under orthogonality constraints [14]. In statistics, some basic problems like estimating the shape of an elliptical distribution [35, 16] or estimation matrix normal models [29, 4] are best viewed through the lens of geodesic convexity.

In recent years there has been significant effort to adapt the key tools and ideas in convex optimization to the Riemannian setting. This includes giving new deterministic [37], stochastic [21, 30], variance-reduced [25, 36], projection-free [34], adaptive [20] and saddle-point escaping [12, 27] first-order methods. Many new ingredients are needed because the traditional analyses in the Euclidean setting rely on the linear structure. Still, one of the key challenges has remained elusive thus far:

> *Is there a Nesterov-like accelerated gradient method for geodesically convex functions on a Riemannian manifold?*

This question is particularly natural in settings where the curvature is non-positive, since it inherits many useful properties of Euclidean space such as having unique geodesics between any pair of

---

*Department of Mathematics, Massachusetts Institute of Technology. Email: `luh@mit.edu`. This work was supported in part by a Fannie and John Hertz Foundation Fellowship.

†Department of Mathematics, Massachusetts Institute of Technology. Email: `moitra@mit.edu`. This work was supported in part by a Microsoft Trustworthy AI Grant, NSF CAREER Award CCF-1453261, NSF Large CCF1565235, a David and Lucile Packard Fellowship, an Alfred P. Sloan Fellowship and an ONR Young Investigator Award.

points. There has been notable partial progress. Zhang and Sra [38] were among the first to clearly articulate this question. They gave a method that achieves Nesterov-like acceleration *if you start sufficiently close to the optimum*. Since then, the aim has been to develop methods that achieve *global* acceleration. Ahn and Sra [2] gave a partial answer by giving a method that converges strictly faster than gradient descent and eventually accelerates. Martínez-Rubio [23] gave a method that achieves global acceleration but at the expense of having hidden constants that depend exponentially on the diameter of the space.

In this work, our main contribution is to dash these hopes by showing that acceleration is impossible even in the simplest of settings where we want to minimize a smooth and strongly geodesically convex function over the hyperbolic plane. Our proof assumes that the gradient oracle returns an answer that has just an exponentially small amount of noise. In comparison, in the Euclidean setting it is possible to achieve Nesterov-like acceleration with an inverse polynomial amount of noise. Of course, in realistic settings some amount of noise is usually unavoidable.

**Theorem 1.** *Given access to a $\delta$-noisy gradient oracle, any algorithm for finding a point within distance $r/5$ of the minimum of a 1-strongly convex and $O(r)$-smooth function in the hyperbolic plane that succeeds with probability at least $2/3$ must make at least*

$$\Omega\left(\frac{r}{\log r + \log 1/\delta}\right)$$

*queries in expectation. Here $r$ is a bound on how far the optimum is from the origin.*

**Comment 1.** While it may at first seem like a limitation to restrict to functions whose condition number depends on the radius, we show in Section 5 that in the hyperbolic plane this is inevitable in the sense that *every* geodesically convex function has a condition number that is at least linear in the radius.

See Theorem 3 for the full version. The key intuition is short and simple: *In negatively curved spaces, the volume of a ball grows so fast that information about the past gradients is not useful in the future.* Indeed for discrete approximations to the hyperbolic plane, like a 4-regular tree, it is not hard to make this intuition precise. This intuition also helps clarify why existing acceleration results need to assume that you are already within a constant neighborhood of the optimum or depend badly on the radius.

It is more challenging to reason about general algorithms that can make queries anywhere they like and not just at a discrete set of locations. The proof of our main result is based on an abstraction in terms of a game where a player wants to find a hidden item in a set and can make queries to a noisy oracle. We prove an information-theoretic lower bound for this game, and then connect it back to convex optimization over curved spaces. The key point is that balls in hyperbolic space grow exponentially with their radius, and this is in turn related to the size of the set in our game. We believe that our query lower bounds for the noisy oracle game can be more broadly applicable, even beyond the realm of optimization.

**Remark 1.** The natural open question in our work is to prove similar lower bounds when we are given an exact gradient oracle. However any such algorithm that depends on exact gradient computations, and fails with even an exponentially small amount of noise, would seem to be rather brittle to issues of practical concern, like rounding due to machine precision.

This paper is structured as follows: in Section 2 we introduce the reader to the counterintuitive properties of the hyperbolic plane. Section 3 defines the noisy gradient problem and phrases it as a more general 'noisy query game'. Section 4 proves our crucial lower bound on general noisy query problems, and then applies it to prove our no-go theorem about acceleration in curved spaces.

## 2 The Hyperbolic Plane

This section serves as an introduction to the hyperbolic plane $\mathbb{H}^2$, establishing the important facts we use for our proof as well as intuition for our main result. The first and most important fact about the hyperbolic plane is that it is large:

**Fact 1.** [10] The circumference and area of a hyperbolic circle are both exponential in its radius.

For illustration, consider a tiling of the hyperbolic plane with congruent equilateral pentagons. As you can see, the number of pentagons at distance $r$ from the origin grows exponentially in $r$.

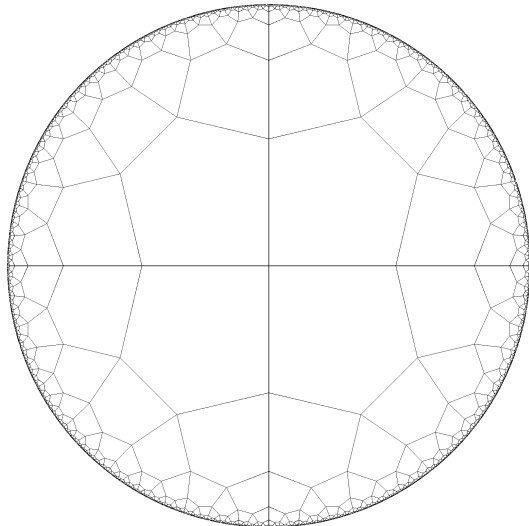

This gives intuition for our result: When attempting to minimize a function whose minimum lies somewhere within a ball of radius $r$, the hyperbolic plane forces you to search over a much larger area than any fixed dimension of Euclidean space would. This inherently makes it harder to exploit information from past queries about the function value and gradient, as you can in acceleration in the Euclidean case, when we are interested in reaching a point far away from the origin.

## 2.1  Convexity

**Definition 1.** [32, 1, 31] A function from a manifold (here, the hyperbolic plane $\mathbb{H}^2$) to $\mathbb{R}$ is geodesically convex if it is convex along any geodesic. This is equivalent to the Hessian of the function having nonnegative eigenvalues. We say that a function is $\alpha$-strongly convex if the eigenvalues of its Hessian are bounded below by $\alpha$. Moreover it is $\beta$-smooth if the eigenvalues of its Hessian are bounded above by $\beta$.

Note that even though a function $f : \mathbb{H}^2 \to \mathbb{R}$ may live in the hyperbolic plane, its gradient at any point, denoted by $\nabla f(x)$, lives in a space isomorphic to $\mathbb{R}^2$. And similarly its Hessian at that point is an ordinary $2 \times 2$ matrix. This is because the gradient and Hessian are local properties of a function, and locally the hyperbolic plane looks Euclidean. This can be made formal through the concept of tangent spaces, but will not be needed for our purposes.

In this paper we will prove lower bounds for minimizing arguably the simplest geodesically convex function, the distance squared function:

**Fact 2.** In the hyperbolic plane, the distance squared function $x \mapsto d(x, x^\star)^2$ is geodesically convex and its minimum is $x^\star$. At distance $r$ from $x^\star$, this function is 1-strongly convex and $(r/\tanh r)$-smooth.

The strong convexity and smoothness come from the formula for its Hessian given in [15].

## 2.2  Why Pirates Don't Search for Treasure in the Hyperbolic Plane

The purpose of this subsection is to provide intuition for our main theorem and is not required to understand our results. Imagine a pirate who has buried treasure in the desert somewhere at distance 100 away from her. She does not remember exactly where the treasure is, but is in possession of a compass which points towards it. The compass' reading has error, though: on the order of $10^{-16}$ degrees. (This setting is analogous to an algorithm able to make noisy queries to the gradient of some function.) In the Euclidean plane, the pirate could easily find the treasure: take a compass bearing, walk 100 steps, and dig. An error of $10^{-16}$ degrees would literally be subatomic.

However, if the pirate attempted this strategy in the hyperbolic plane, she would end up at distance just over 190 from her treasure. She would have started to walk away from the treasure after just a few steps! (Specifically, a constant number of steps that scales with $\log(1/10^{-16})$.) Therefore, she

would have to repeatedly look at her compass every few steps in order to make progress towards the treasure. This is why gradient descent has only linear convergence in hyperbolic spaces. Of course, this falls short of explaining why no algorithm converges faster. One of the authors (though we won't say who) is indebted to the video game HyperRogue [22] for giving intuition about the hyperbolic plane. One level of this game features a similar scenario with pirates and compasses.

## 3 Optimization with a Noisy Gradient Oracle

In this section we define the main model we will be interested in. Moreover we recall convergence bounds in the Euclidean case, particularly those that continue to hold in the presence of a small amount of noise, as a point of comparison.

**Definition 2.** The radius-$r$ gradient optimization model is as follows: There is an unknown differentiable function $f$ whose minimum is within distance $r$ of the origin. An algorithm may query points $x$ within distance $1000r$ of the origin.[3] Upon querying $x$, the algorithm learns $f(x)$ and the gradient $\nabla f(x)$.

**Definition 3.** In the noisy version of the model, instead of learning the exact values of $f(x)$ and $\nabla f(x)$, the algorithm receives $f(x) + z_1$ and $\nabla f(x) + z_2$ where $z_1$ and $z_2$ are noise. We do not require the noise to be of a specific form such as Gaussian, uniform, etc – we only require that the noise terms for different queries are independent.

**Definition 4.** We say noise is *c-non-concentrated* if on any query, the probability distribution function of the noise is everywhere bounded above by $c$. We say noise is *C-precise* if the noise term never has magnitude larger than $C$.

In Euclidean space, a small amount of noise does not preclude acceleration. As a point of comparison, we restate the key result (Theorem 7) from [13]:

**Theorem 2** (from [13]). *There is an algorithm that, given $C$-precise noisy oracle access to an $L$-smooth $\mu$-strongly convex function $f$ and its gradient, along with a starting point at distance $d$ from the minimum of $f$, outputs after $k$ oracle queries a point $x_k$ such that*

$$f(x_k) - \min f \leq O\left(Ld \cdot \min\left(1/k^2, \exp(-k/2\sqrt{\mu/L})\right) + \min\left(k \cdot poly(C), \sqrt{L/\mu}\right)\right)$$

(The bound in the original paper is more precise, making big-O constants explicit and using a more refined notion of precision.) This theorem implies that accelerated gradient descent works in Euclidean space even in the presence of noise, provided that the noise has magnitude at most some inverse polynomial in $r$. Contrast our main result: in hyperbolic space, accelerated gradient descent is impossible even with *exponentially* small noise.

## 4 The Noisy Gradient Task in the Hyperbolic Plane

Recall, we will be interested in the simplest example of a convex function in the hyperbolic plane: $f(x) := \text{dist}(x, x^\star)^2$, where $x^\star$ is some point at distance $r$ from the origin. Finding the minimum of $f$ is equivalent to locating $x^\star$. Without noise, an algorithm could locate $x^\star$ exactly in *one* query, because any gradient is guaranteed to both point exactly at $x^\star$ and indicate the distance to $x^\star$. What about with noisy gradients?

Within the hyperbolic of radius $r$ centered at the origin, our function $f$ is $O(r)$-smooth and 1-strongly convex. (As an aside, in Theorem 6 we show that for any $\beta$-smooth and $\alpha$-strongly convex function in the hyperbolic disk we must have $\beta/\alpha = \Omega(r)$). If Nesterov-like acceleration in the hyperbolic plane were possible we should be able to locate $x^\star$ to within distance 1 in time $O(\sqrt{r})$. Unfortunately, as we will show, this task is impossible. Even worse, it is impossible to get any polynomial factor speedup in the convergence.

---

[3]The number 1000 is arbitrary and only affects constants inside big-O notation. The reason for this assumption is as follows: if an algorithm were to query a point $x$ superexponentially far away from the origin and learn $f(x) + $ noise, then $f(x)$ would be so large that the noise term could be disregarded completely. Thus, removing this assumption would require a more refined noise model, such as multiplicative noise.

**Theorem 3.** *In the radius-$r$ noisy gradient optimization model in the hyperbolic plane, if queries receive noisy answers with $c$-non-concentrated $C$-precise noise, then any algorithm that can find a point within distance $r/5$ of the minimum of the function at succeeds with probability at least $2/3$ must make at least*

$$\Omega\left(\frac{r}{\log r + \log C + \log c}\right)$$

*queries. This is true even if the function is guaranteed to be $1$-strongly convex and $O(r)$-smooth at every point within distance $r$ from the origin.*

To prove this result, we will generalize the noisy gradient model to any setting in which an agent makes queries and receives probabilistic answers over a discrete set of possibilities. In this general setting, we will prove a lower bound on the number of queries needed to determine the state of the world.

## 4.1 Noisy Query Games

**Definition 5.** A *noisy query game* is a tuple $(n, Q, \mathcal{X}, f)$, with which the following one-player game is played: A secret number $i^\star$ is chosen uniformly at random from $\{1, 2, \ldots, n\}$. The player's goal is to determine $i^\star$. To do so, the player may make a query $q \in Q$ and receive an observation $X \in \mathcal{X}$. The observation is sampled using some probability distribution function $f_{q, i^\star}(x)$. The player wins when they can guess $i^\star$ with probability at least $2/3$.

We remark that for us $\mathcal{X}$ will be a region in Euclidean space and we will use $|\mathcal{X}|$ to denote its volume. The noisy game broadly seems to be a natural model for a class of noisy learning tasks. In particular, it generalizes the noisy gradient task in the hyperbolic plane, as we show in the following comment:

**Comment 2.** Consider the noisy gradient task in which we place $n = e^{\Theta(r)}$ points equally in a circle of radius $r$ in the hyperbolic plane, so that the points are distance $\geq r/2$ apart. (This is possible because circles are exponentially large – see [10], page 92 – so greedily picking points one at a time and removing a ball of radius $r/2$ around each runs out of volume only after exponentially many steps.) The secret number $i^\star$ corresponds to one of these points $x^\star$. Define the function $f(x) = \text{dist}(x, x^\star)^2$ whose optimum is $x^\star$. The player makes queries in $Q :=$ a region in the hyperbolic plane with radius $O(r)$, and receives noisy gradient observations. Now the player can win if and only if they can locate the optimum of $f(x)$, among the discrete set of possibilities, with probability at least $2/3$.

We now state our lower bound for noisy query games:

**Theorem 4.** *In a noisy query game $(n, Q, \mathcal{X}, f)$, suppose the noise is $c$-non-concentrated, i.e. all probability distribution functions $f_{q, i^\star}$ are everywhere bounded above by some constant $c$. Then for the player to be able to guess $i^\star$ with probability at least $2/3$, the player must make at least $\Omega(\frac{\log n}{\log(c|\mathcal{X}|)})$ queries.*

Theorem 4 is difficult to prove directly, because the player's knowledge is a posterior distribution over the options $\{1, 2, \ldots, n\}$ which can change in complicated ways. To overcome this obstacle and prove Theorem 4, we will define an easier 'transparent' version of the noisy query game, where the player's knowledge is a subset of $\{1, 2, \ldots, n\}$ representing which options could possibly be the correct one. Then we will show that even in the easier version of the game, the player needs many queries in expectation to succeed.

## 4.2 The Transparent Noisy Query Game

In a noisy query game, observations are sampled from probability distribution functions $f_{q,i}$ on a space $\mathcal{X}$. One way to sample an observation from $f_{q,i}$ is to sample uniformly from the region under its graph. Let $G_{q,i}$ denote this region. Note that the volume of $G_{q,i}$ must be 1, because probabilities always sum to 1. In the transparent noisy query game, we answer a query $q$ by telling the player a point $(x, y)$ uniformly sampled from the graph region $G_{q,i^\star}$. In the normal query game the player only learns $x$, so the normal version can only be harder.

The key question is: How does the player's knowledge update when she receives an observation $(x, y)$? For any option $i$ whose graph area $G_{q,i}$ does not include the point $(x, y)$, the player learns

that $i$ cannot possibly be correct. For the rest of the options, the player learns nothing. This is because observations are sampled uniformly and each $G_{q,i}$ has unit area, so by Bayes' rule the player's posterior on $i^\star$ remains uniform over all remaining options. (Here we have assumed that the prior is uniform at the beginning.)

Indeed, this convenient property is the reason we defined this transparent version: It allows us to easily analyze the player's progress by tracking only the number of remaining possible options, rather than the messy details of what happens to the posterior distribution.

**Lemma 5.** *Suppose all the $f_{q,i}$ are $c$-non-concentrated distributions. Then in the transparent noisy query game, a query decreases the logarithm of the number of possible remaining options by at most $\log(c|\mathcal{X}|)$ in expectation.*

*Proof.* Let $m$ be the number of options remaining before the query. For convenience, use the notation $N(x, y)$ for the number of graph areas $G_{q,i}$, among the $m$ remaining options, that contain $(x, y)$. If the player receives the query result $(x, y)$, they would be left with $N(x, y)$ remaining options. So after the query, the expected number of options remaining is

$$\mathbb{E}_{i^\star}\left[\int_{G_{q,i^\star}} \log\left(N(x, y)\right)\,dxdy\right],$$

where the expectation is taken uniformly at random from among the $m$ remaining options. Moving the expectation inside the integral sign and using the assumption that all graph areas are contained within $\mathcal{X} \times [0, c]$, we get that the above expectation is equal to:

$$\int_{\mathcal{X}\times[0,c]} \frac{N(x, y)}{m} \log\left(N(x, y)\right) dxdy$$

Since each graph has area 1, the integral $\int_{\mathcal{X}\times[0,c]} N(x, y)$ is $m$. So by Jensen's inequality, subject to this restriction, the above quantity is minimized when $N$ is constant over the entire domain $\mathcal{X} \times [0, c]$. The minimum value is

$$\int_{\mathcal{X}\times[0,c]} \frac{m/c|\mathcal{X}|}{m} \log\left(m/c|\mathcal{X}|\right) dxdy = \log\left(m/c|\mathcal{X}|\right) = \log m - \log\left(c|\mathcal{X}|\right)$$

So the expectation of the logarithm of the number of possible options left decreases by at most $\log(c|\mathcal{X}|)$, as desired. $\square$

Now we can prove our main lower bound for the noisy query game:

*Proof of Theorem 4.* Let $n_i$ denote the number of possible remaining options after $i$ steps. Thus we have $n_0 = n$. Now let $X$ be a random variable that represents the cumulative progress the algorithm has made. In particular let

$$X = \sum_{i=1}^{T} \log n_{i-1} - \log n_i$$

Applying Lemma 5 and Markov's bound we have that $X \leq 3\mathbb{E}[X]$ with probability at least $2/3$. If the algorithm succeeds at being able to determine $i^\star$ after $T$ steps we must have $n_T = 1$. Putting everything together we have

$$0 = \log n_T = \log n - X \geq \log n - 3|T|\log(c|\mathcal{X}|)$$

and rearranging completes the proof. $\square$

### 4.3 Proof of the Main Theorem

Our main result now follows easily from the machinery of noisy query games:

*Proof of Theorem 3.* Suppose we want to minimize the function $f(x) = \text{dist}(x, x^\star)^2$. This function is 1-strongly convex and $2r$-smooth within distance $r$ of the origin. (As mentioned earlier, [15] shows the eigenvalues of the Hessian are 1 and $r/\tanh r \leq r + 1$.) First we apply the reduction in Comment 2 so that we have $n = e^{\Theta(r)}$ points with pairwise distance at least $r/2$. Moreover $x^\star$ is among them and corresponds to the secret number $i^\star$ in the noisy query game.

In the setting of Theorem 3, a player makes queries within a certain region of the hyperbolic plane, and learns the (noisy) function value and gradient at their query point. They are tasked with finding a point within distance $r/5$ of $x^\star$. Because the $n$ points have pairwise distance at least $r/2$, doing so requires figuring out which of the $n$ points is $x^\star$. So the player must win the query game, which by Theorem 4, takes at least $\frac{\log n}{\log(|X|c)}$ queries.

We picked $n = e^{\Theta(r)}$ above, and the value of $c$ is stated in Theorem 3's assumptions. But what is $|\mathcal{X}|$, i.e. the volume containing all query answers? Since the player's queries are restricted to a region in the hyperbolic plane of radius $O(r)$, the true answer to their query is a function value in the interval $[0, O(r^2)]$ along with a gradient in the disk $B(0, O(r)) \subseteq \mathbb{R}^2$. (Recall from Section 2.1 that the gradient lives in $\mathbb{R}^2$, not the hyperbolic plane.) By assumption, the noise causes error at most $C$, so the observed query answer must lie in

$$[-C, O(r^2) + C] \times B(0, O(r) + C) \subset \mathbb{R}^3$$

This is a compact set whose volume is a polynomial in $r$ and $C$. In particular we have $|\mathcal{X}| \leq O(r^4 C^3)$. Therefore overall the player needs at least

$$\frac{\log n}{\log(|\mathcal{X}|c)} = \frac{r)}{\log(c) + O(\log r + \log C)}$$

queries. This completes the proof. $\qquad\qquad\square$

## 5 Lower Bounds on the Condition Number

In Euclidean space, the function $f(x) = ||x||^2$ is 1-smooth and 1-strongly convex at every point. However, as we will show, in the hyperbolic plane geodesically convex functions always have a condition number that depends on the radius:

**Theorem 6.** *If $f$ is a $\beta$-smooth, $\alpha$-strongly convex function defined in a hyperbolic disk of radius $r$, then $\beta/\alpha \geq \Omega(r)$.*

*Proof.* First we will give the intuition for the proof. Consider a geodesic that dips a distance of 1 into the disk of radius $r$ (see the picture below). On the one hand, due to $\alpha$-strong convexity, the value of $f$ must vary a large amount along this geodesic. But on the other hand, this geodesic is short, so by $\beta$-smoothness $f$ cannot vary much. These two properties will give us a lower bound on the condition number.

Now we proceed to the formal proof. Of all points at distance $r - 1$ from the center of the disk, let $x$ be one at which $f$ is minimal. Without loss of generality suppose that $f = 0$ at the center of the disk. By convexity and the minimality of $x$, we deduce that

$$f(y) \geq \frac{r}{r-1} f(x)$$

for all $y$ on the circumference of the disk. By $\alpha$-strong convexity, $f(x) \geq \Omega(\alpha r^2)$. Now draw a geodesic through $x$, as pictured, perpendicular to the geodesic between the center of the disk and $x$. This geodesic intersects the disk at two points $y$ and $y'$. By $\beta$-smoothness,

$$\frac{1}{2}(f(y) + f(y')) \leq f(x) + O(\beta d(y, y')^2)$$

Finally, in hyperbolic geometry, the distance $d(y, y')$ is $O(1)$. See Lemma 7 for an explicit calculation justifying this. Finally combining the three inequalities in the previous paragraph gives $\beta/\alpha \geq \Omega(r)$, as desired. $\qquad\qquad\square$

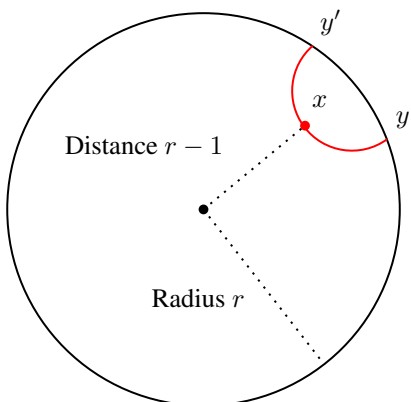

**Lemma 7.** *A hyperbolic triangle with two sides of length $r$, whose altitude between those sides has length $r - 1$, has a third side of length $O(1)$.*

*Proof.* This is a straightforward calculation using formulas from [19]. Let $c$ denote the length of the third side and $A$ denote the measure of either angle adjacent to side $c$. (Those angles are equal because the triangle is isoceles.) The hyperbolic law of cosines from [19] gives $\cos A = \frac{(-1 + \cosh c) \cosh r}{\sinh c \sinh r}$. The altitude length formula gives $\sin A = \frac{\sinh(r-1)}{\sinh r}$. Using $1 - \sin^2 = \cos^2$ gives:

$$1 - \frac{\sinh^2(r-1)}{\sinh^2 r} = \frac{(-1 + \cosh c)^2 \cosh^2 r}{\sinh^2 c \sinh^2 r}$$

Rearranging the above expression, we get:

$$\left( 1 - \frac{\sinh^2(r-1)}{\sinh^2 r} \right) \frac{\sinh^2 r}{\cosh^2 r} = \frac{(-1 + \cosh c)^2}{\sinh^2 c}$$

As $r \to \infty$ the left-hand side tends to $1 - \frac{1}{e^2} \approx 0.86$. Then solving the right-hand side for $c$ gives a solution, unique in the reals, that is approximately 3.31, which is $O(1)$ as desired. $\qquad \square$

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
