# OpenReview forum: "A No-go Theorem for Robust Acceleration in the Hyperbolic Plane"
_NeurIPS.cc/2021/Conference — NeurIPS 2021 Poster_

### Official Review · Reviewer_TFi3 · 2021-07-08

**Rating:** 4
**Confidence:** 3

**Summary:**

The paper presents a lower bound on the complexity of a noisy gradient oracle for minimizing a strongly convex real-valued function defined on the hyperbolic plane. This gives answers regarding the acceleration phenomenon on Riemannian manifolds and limitations of previous works. A lower bound on the condition number of a strongly convex function in the hyperbolic plane is also provided.

**Limitations And Societal Impact:**

The work is theoretical and has no negative societal impact. The authors discuss the theoretical limitations of the paper.

**Main Review:**

Originality: The main idea and techniques presented in this paper surprised me by their originality. I have never seen this kind of arguments applied in deriving lower bounds for optimization, so, to the best of my knowledge this paper is quite novel. The related work is cited properly, there are 2-3 papers in Riemannian acceleration that could be included and discussed though.

Quality: The paper is a complete work presenting a lower bound in the noisy setting. As the authors mention, the only limitation of this work is that does not tackle the exact gradient oracle setting (without noise). In the noisy setting, theorem 3 implies that for minimizing a $1$-strongly convex and $O(r)$-smooth function $\Omega(r/\log(r))$ iterations are required, which is strictly bigger than $O(\sqrt{r})$ that an accelerated method would need if $r$ is sufficiently large (but still compatible with gradient descent). The authors prove also in theorem 6 that the assumption of a function with $O(r)$ condition number in the hyperbolic plane is not restricting. However, there are many parts of the paper that seem problematic to me:

1. The constant $c$ which bounds the distribution should be between $0$ and $1$ and potentially $c \rightarrow 0$, in which case $\log(c) \rightarrow -\infty$. How is this possible to happen in Theorem 3? If you do the same in Lemma 5, it means that you reduce the options by a negative number. You can still put $c=1$ and vanish $\log(c)$ term though. In any case, I do not understand why this $c$-bound on the distribution is essential for the analysis.

2. When $C \rightarrow 0$ the same as above happens with the $\log(C)$ term, which is weird since the noise is assumed to be small. I think there is a mistake in 234, namely the volume of $\mathcal{X}$ is not $O(r^4 C^3)$, but $O(r^4+2 C r^2+2 r^3 C +4 r C^2+r^2 C^2 +2 C^3)$ (if I did the calculation correctly). Thus, when $C=0$ the analysis seems to be valid without the $\log(C)$ term in the lower bound. Given that point I cannot understand what is the role of noise after all. Would be happy if the authors could indicate where exactly introducing noice is needed in the analysis.

3. It is a bit surprising that the result is based on the squared distance function, which is a very easy-to-minimize one. I see that this is comfortable for being able to compute the condition number, but is there any other point that this specific form for $f$ is needed?

4. In comment 2, I don't understand how one can put exponentially many points in a circle with mutual distance $r/2$, when the circumference is $O(e^r)$. I think you can put exponential amount of points only in distance $O(1)$, do I miss something? I don't think that $O(1)$ distance change any result though, the authors themselves write in 152 that the minimization goes until accuracy $1$ and then in theorem 3, until $r/5$, I cannot see why this changes.

5. I understand that given a bound like in Theorem 3, in order to achieve acceleration we must initialize close to the minimizer and stay always in such a neighborhood ($r$ must be small). I don't understand how that explains also the exponential dependence on the domain in [23]. Can you explain? Could you also comment on the relantionship of your result with the one in [Yuanyuan Liu, Fanhua Shang, James Cheng, Hong
Cheng, and Licheng Jiao. Accelerated first-order
methods for geodesically convex optimization on riemannian manifolds] (neurips 2017) ?

Minor issues: i) The function $dist(x,x^*)^2$ is $2$-strongly convex and $2 \frac{r}{tanh(r)}$-smooth, you can multiply with $0.5$ to get an $1$-strongly convex function. ii) In which point of the proofs the probability $2/3$ arise? iii) End of 234 there is an extra parenthesis.

Clarity: The paper is well-written and the main results clearly stated. There are points though that more explanation could be added, see my previous comments. I particularly like the intuitive explanation in section 2.2.

Significance: Given that my previous points are addressed (because in the current form there are points that do not make sense to me), the paper proves that adding noise in the oracle used for optimization of a cost function in the hyperbolic plane is much worse than doing the same in the Euclidean space. This is because the circumference of a circle in the hyperbolic plane is $O(sinh(r))=O(e^r)$ and the area is $O(e^{2r})$. To the best of my understanding this problem is orthogonal to the very problem of acceleration with a non-noisy oracle. For instance, in the sphere the behaviour of circles is exactly the opposite, i.e. the circumference is $O(sin(r))$, thus I would expect that adding noise to an oracle for optimizing in the spherical case to yield to better results than the Euclidean case. In [38] for instance, the effect of curvature seems to be symmetric between the positive and negative case. This is the case also in  [Foivos Alimisis, Antonio Orvieto, Gary Bécigneul, Aurelien Lucchi, Momentum Improves Optimization on Riemannian Manifolds]. I do believe that optimization itself is more difficult in lower curvature, but so far the main problem seems not to be that, but rather non-linearity. Non-linearity on the other hand does not seem to affect directly the noise as discussed previously with the example of the sphere.

Overall, I believe that these are good to know results regarding how noise behaves in negative curvature, but I don't think that are directly related to the problem of acceleration with non-noisy oracle.

Given the above, I think that there are many issues for the authors to address and vote for rejection. I am willing to change my evaluation though if my main concerns are answered in the rebuttal.










**Time Spent Reviewing:**

16

---

> ### Author Response · Authors · 2021-08-09
> **Many points of confusion**
>
> Thanks for your careful reading of our paper, however there appear to be many points of confusion.
>
> First, we disagree with your assessment of the significance of our results. In the Euclidean case, the fact that accelerated methods can tolerate a small amount of noise is not just a footnote. Most practical applications of these methods compute gradients up to machine precision, suffer from roundoff errors and other sorts of noise. If these methods broke down outside of their idealized setup, they would not be nearly as useful or have such wide ranging applications. So, yes, while it is still possible that non-robust acceleration can be achieved for negatively curved manifolds, our results place serious limitations on how useful those methods could be. Computing exact gradients is a challenging task that requires having exact knowledge of the function being optimized.
>
> The constant $c$ does not need to be between zero and one. It is an upper bound on the p.d.f. of the additive stochastic noise. Since $C$ is a bound on its support, we must have that $C \geq c$, otherwise it would not be a properly normalized distribution. Thus $\log C + \log c \geq 0$.
>
> Thanks for raising the issue about $C$. We should have written that $C \geq 1$. It is an upper bound on the support of the noise, and the main purpose of our theorem is not to show that we get better bounds as $C$ decreases (e.g. it goes to zero) but to show a $r / \log r$ type bound that holds even for moderately large values of $C$. This is why the estimate $O(r^4 C^3)$ bound holds. The detailed version you wrote out is correct, but for our purposes the $O(r^4 C^3)$ is loose but good enough.
>
> The reason we use the squared distance function is because its gradients are easy to work with in conjunction with our noisy query game. It is possible that other functions would work too.
>
> In regards to Comment 2, the argument works based on the volumes. The volume of the hyperbolic disk of radius $r$ is $\Theta(e^{\alpha r})$ for some $\alpha > 0$. See reference [10]. Then when we choose an arbitrary point in the hyperbolic disk (not necessarily on the boundary), we remove everything within distance $r/2$ of it. The volume of the remaining points decreases by at most $\Theta(e^{\alpha/2 r})$. Since there is still volume left over, we can pick another point and repeat the process. Thus we select at least $\Theta(e^{\alpha/2 r})$, each of which is at distance at least $r/2$ from all the other points we have selected, by construction. For intuition, you can think of the complete $4$-ary tree of depth r. This is a common discrete model for the hyperbolic plane, and again you can select an exponential number of nodes that are distance at least $r/2$ from each other.
>
> In regards to reference [23], we do not claim to show the exponential dependence on $r$ is necessary. In fact we suspect that it is not. After all, one can use a first order method without acceleration to get near to the optimum in $O(r)$ steps and then appeal to some of the known local acceleration results. For the paper of Liu et al., their method is an implicit iterative process. In each step, they need to solve a complex equation (4) and no bounds are given on the number of oracle calls needed, except in special cases like averaging real p.s.d. matrices.
>
> Finally, we do not understand the comment about the work of Alimisis et al. While it is true that their bounds are symmetric in the positive and negative curvature cases, the bounds themselves require the sectional curvature to be bounded in terms of the diameter in order to achieve acceleration. Thus the further you are from the optimum, the flatter the manifold must be (and then much of the intuition carries over from the local acceleration cases). The point of our work is that when the curvature is independent of the diameter, the volume of a negatively curved manifold grows too fast and past gradients are not as useful in trimming down the search space, in contrast to the Euclidean case.

---

> > ### Comment · Reviewer_TFi3 · 2021-08-16
> > **Rephrasing of some concerns**
> >
> > I thank the authors for clarifying many of the points raised, I understand that $c$ and $C$ are upper bounds and can be set as large as we wish. However, I am still confused with the role of noise and would like to rephrase some questions:
> >
> > 1. Why the proof of Lemma 5 does not work for $c=0$? Setting $c=0$, we get an integral over the domain $\mathcal{X} \approx \mathcal{X} \times \lbrace 0 \rbrace$ instead of $ \mathcal{X} \times [0,c] $ and the quantity to be minimized is minimized again when $N$ is constant over $\mathcal{X}$, i.e. $N = m/ \mid \mathcal{X} \mid $. Then, we can rewrite the same argument and get $ \int_{\mathcal{X}} \frac{m/ \mid \mathcal{X} \mid}{m} \log (m/\mid \mathcal{X} \mid) dx dy = \log m - \log \mathcal{X}$. Thus, $c$ just goes away.
> >
> > 2. Why the proof of main theorem (section 4.3) does not work with $C=0$? For $C=0$, the domain $[-C, O(r^2)+C] \times B(0, O(r)+C)$ becomes $[0,O(r^2)] \times B(0,O(r))$, which has volume $O(r^4)$.
> >
> > Thus, as far as I understand, we get exactly the same result without the terms $\log c, \log C$. What do I miss?

---

> > > ### Author Response · Authors · 2021-08-17
> > > **Thanks for the questions**
> > >
> > > Thanks for the questions. The answer to all of them is basically because the distribution on noise must be properly normalized.
> > >
> > > 1. Recall that the definition of $c$-non-concentrated is that the probability density function of the noise is everywhere bounded above by $c$. So you cannot set $c = 0$ because then it could not be a distribution. Moreover you cannot hold everything else fixed and make $c$ arbitrarily small either, because if it is smaller than $1|/\mathcal{X}|$ then again you cannot have a distribution.
> > >
> > > 2. Recall that the definition of $C$-precise is that the noise must have magnitude larger than $C$. Thus if we set $C = 0$ then we must have $c$ is infinite. In the proof of Theorem 3, we assumed $C \geq 1$ and if you instead set $C = 0$ then $|\mathcal{X}| \leq O(r^4)$. Thus the lower bound on the number of queries $\log n/ \log (|\mathcal{X}| c)$ becomes vacuous because the denominator is infinity.

---

### Official Review · Reviewer_XUX6 · 2021-07-09

**Rating:** 5
**Confidence:** 3

**Summary:**

This work shows that an inexact accelerated gradient method for geodesically convex functions on a Riemannian manifold might not exist by showing a lower bound for an optimization problem in the Hyperbolic 2-space.

The proof is based on reduction to playing a game, in which there are $n$ probability distribution functions, and the player can make a query and receive an observation sampled from a specific distribution function and need to figure out which function generates the samples.


**Limitations And Societal Impact:**

The authors should addressed the limitation in this rebuttal.

**Main Review:**



**1. (Significance)**

It is unclear to me how strong/applicable the negative result is. Specifically, does it say anything on problems like matrix completion [9, 28, 33], dictionary learning [11, 26], robust subspace recovery [39], mixture models [18]? Is acceleration not possible in these problems?  The authors might want to discuss the connections.



**2. (Proof)**

I am not sure if the proof is correct and I have some questions.

2.1 (Between line 234-235)

It looks to me that if the underlying optimization algorithm can only query function values but cannot get gradients, then the proof of this paper still applies and the number of the query needed would also be $\Omega(r)$, where $r$ is the condition number.  From the equation right above line 235, it looks like the linear dependency on the condition number $r$ is solely due to the number of points is $n = \exp( \Theta(r) )$ (right below line 234) in the Hyperbolic 2-space. That is, the $\Omega(r)$ complexity does not depend on whether the algorithm has the gradient information or not, which means that the gradient information does not help!? Something wrong?


2.2 (Intuition of the reduction)

In this rebuttal, can the authors provide an intuition why the proof can be reduced to playing a game?



**3. (Clarity)**

This paper is not self-contained from my perspective. For example, the authors should provide derivations of Fact 2. Also, the argument on line 170-175 does not appear to be rigorous. It would be helpful if the authors can provide the details in the supplementary and/or in this rebuttal.


**Time Spent Reviewing:**

6

---

> ### Author Response · Authors · 2021-08-09
> **Thanks for your review**
>
> Thanks for your careful reading of our paper.
>
> First, we would like to emphasize that the problem of minimizing geodesically convex functions on curved spaces is well studied in its own right. One of the main open problems, when porting the toolkit from standard optimization to the curved case, is to design first order methods that achieve acceleration. We show that any such method must be inherently non-robust in the sense that its acceleration would evaporate if the gradients it receives are allowed to have even an exponentially small amount of noise. This puts serious limitations on what researchers in the area were aiming to do because in most practical scenarios we only compute gradients up to machine precision, and incur roundoff errors. We are not aware of any connections between our problem and the other problems you suggested.
>
> In terms of correctness, we would be more than happy to clarify any points of confusion. In your hypothetical example where an algorithm can only make function queries and not gradient queries, why is it surprising that the lower bound of $\Omega(r/\log r)$ still holds? Allowing the algorithm less power, to only make function queries and not gradient queries, only makes it easier to prove lower bounds. This does not mean that gradients do not help, as you suggest.
>
> The proof of Fact 2 depends on computing the Hessian of a function in the hyperbolic plane. There is actually a fair amount of background in differential geometry we would need to add just to define this properly and in the standard way. However there is a simple way to provide intuition for why it ought to be true: There is a heuristic model for the hyperbolic plane that makes things discrete. Take a $4$-ary tree of depth r, rooted at the origin. Then the function $distance(x, x^*)^2$, where $x^*$ is a leaf node, can be seen to be geodesically convex. Here the distance is the unweighted graph distance on the tree and the geodesics between pairs of nodes are the (unique) shortest paths.
>
> The argument in Lines 170-175 is indeed rigorous. It is a standard version of a packing and covering argument. To elaborate, the volume (area) of a circle of radius r in the hyperbolic plane is $\Theta(e^{\alpha r})$ for some $\alpha > 0$. This is shown in reference [10], on page 92. Now if we pick any point in the circle, and remove all points within distance $r/2$, we would remove at most $\Theta(e^{\alpha/2 r})$ volume. Thus since there is still volume left over, we can pick another point that has not yet been removed. This process continues for at least $\Theta(e^{\alpha/2 r})$ steps. And no pair of selected points can be closer than distance $r/2$ because otherwise whichever point we removed first, we would have removed the ball of radius $r/2$ around it and removed the other point too.
>
> Finally, the main intuition behind the reduction to the query game is that (1) We are making the problem easier by restricting the optimum to be among some finite, but exponentially large set, as in Comment 2. (2) If we make a query at some point, and consider two candidate optimal solutions that are far away in the hyperbolic plane, the vectors pointing to them (i.e. the gradients, if either of those two points were the optimal solution) would be pointing in almost the same direction. Thus adding even an exponentially small amount of noise to these directions, we can couple them so that they almost always have the same value.
>
> We will add these clarifications/elaborations to the next version of the paper. We think it will improve the readability, and thank you for pointing them out.

---

### Official Review · Reviewer_mGYV · 2021-07-16

**Rating:** 7
**Confidence:** 3

**Summary:**

The authors show that when optimizing a hyperbolic function using noisy gradients, it is impossible to achieve speedup akin to that of Nesterov's momentum even when the function is geodesically convex. They do this by reformulating the optimization procedure as a query game.

**Ethical Concerns:**

There are no ethical issues.

**Limitations And Societal Impact:**

The authors adequately addressed limitations and potential negative societal impact.

**Main Review:**

Strengths

* The result is interesting and has many application for machine learning on hyperbolic space.
* The proof technique is clever and is an overall nice construction.
* The proofs and constructions are clear and the intuition is well explained and provides good insight.

Weaknesses

* The proof is currently a bit limited in scope. In particular, it may be helpful to touch upon cases where the curvature $K$ of the hyperbolic space is not $-1$. These have seen practical use, but I expect the result to be quite similar.

Verdict

The paper is compact and well structured, and the central result is important both from a theoretical and practical standpoint. I advocate acceptance.



**Time Spent Reviewing:**

2

---

> ### Author Response · Authors · 2021-08-09
> **Thank you for your review**
>
> Thank you for your review, and your enthusiasm for our work!
>
> It would be quite interesting to extend our results to the more general setting of negatively curved manifolds where the curvature is upper bounded by $-1$. There seem to be many technical challenges in doing so. Just to give a flavor of the types of complexities that arise: In the Euclidean case, if I give you the values of a function at a finite set of points, there is a polynomial time algorithm to decide whether or not they can be extended to a convex function. But the corresponding problem in negatively curved manifolds is wide open. It's not even known if the problem is decidable. This makes extending to negatively curved manifolds difficult, because it's not obvious how to generate nice classes of geodesically convex functions.

---

### Official Review · Reviewer_seq7 · 2021-07-16

**Rating:** 7
**Confidence:** 4

**Summary:**

(Here and below, I will write "smooth" in the sense of differentiability, and
"gradient Lipschitz" for "smooth-in-the-optimization-sense")

The authors prove that in the noisy oracle model (so function values and
gradients are noisy, not exact), there is no accelerated Riemannian gradient
descent algorithm (Nesterov-type acceleration) that succeeds whp uniformly over
all manifolds and all geodesically convex functions. To show this, they
study the minimization of a particular smooth geodesically convex function on
the hyperbolic plane $\mathbb{H}^2$, namely the the intrinsic distance to a
fixed point ${x}^{\star}$ (i.e. ${x} \mapsto \mathrm{dist}({x}, {x}^{\star})$),
and prove the lower bound with this function; they also establish that this
function is not pathological among functions on the hyperbolic plane in the
sense that its gradient-Lipschitz constant is of near-minimal size.
The proofs use elementary properties of the hyperbolic plane -- gradient and
Hessian expressions for the distance, and the fact that ball volumes in the
hyperbolic plane are exponential in the radius -- as well as a reduction to a
"noisy query game" problem on which the lower bound is established.


**Limitations And Societal Impact:**

Yes.

**Main Review:**

## Summary wrt decision

The paper is written, referenced, and motivated well, and provides a
noteworthy contribution to a problem that has seen extensive study in the
literature. The authors' contributions will no doubt help to focus future
research in this line: for example, by focusing on acceleration on
nonnegatively curved spaces, or on investigating the open problem of lower
bounds for optimization with exact gradients. My rating just reflects a few
points of confusion listed below that I had while reading the proofs -- I
expect to increase it after these are resolved in the discussion phase.

## Detailed comments

- Because the number of queries we should expect to have if acceleration in the
  hyperbolic plane is possible is only asserted, rather than shown (sentence at
  line 151), I am having some trouble reconciling the "no robust acceleration"
  claim with the guarantee of Theorem 3 -- maybe it would be helpful if this
  was spelled out in detail (i.e. both a precise illustration of this claim in
  line 151 and a precise definition of what is taken to be an accelerated
  method in this context). In particular, is the claim "... Nesterov-like
  acceleration in the hyperbolic plane [is] possible" equivalent to the
  existence of a guarantee like the one in Theorem 2? I think this equation is
  not labeled correctly ($d$ should be the squared distance to the optimum /
  the squared diameter of the domain; I'm looking at Theorem 7 in the cited
  reference [13]). Then if I interpret a formula like this in the context here,
  we have $L = r$ and $d = r^2$, and ignoring the noise term and the local
  linear rate would give a formula $\mathrm{dist}(x_k, x^\star) \leq r^3 /
  k^2$, so to get within distance $1$ wouldn't we expect to need time
  $O(r^{3/2})$? (should it be within distance $r$ instead?) If we're using the
  the local linear rate here to get the prediction, this feels like it might
  not preclude the existence of accelerated methods. If the intention was
  rather to demonstrate that the lower bound proved here is met by a
  non-accelerated gradient descent method (hence ruling out the possibility for
  acceleration to exist), it might be nice to have this discussed in detail as
  well. I am not up-to-date on the latest rates for smooth strongly
  geodesically convex optimization, but if I look at the cited reference [37],
  (Theorem 13 and Theorem 15), it seems like these rates are not necessarily
  inconsistent with the possibility of having an accelerated method (the key is
  that relative to the euclidean setting, these rates may have extra unsavory
  dependences involving $\zeta(\kappa, r) \approx r$ here; I would appreciate
  correction if sharper rates have already been established).
  In general, the possibility of having these "extra constants" appear in the
  rates relative to the euclidean setting makes me feel that it may be hard to
  use a query complexity lower bound to show impossibility of acceleration
  without better upper bounds for vanilla gradient descent: for example, adding
  an "extra $r$" into the numerator as above (corresponding to a curvature term
  like the $\zeta(\kappa, r)$ in [37]) would lead us to expect to need $r$
  queries to get within distance $O(r)$ with an accelerated method, which would
  be consistent with Theorem 3's lower bound. I would appreciate some
  clarification on these points (I apologize if I am missing something
  obvious).
- It would nice to see additional comparisons to some pieces of related work,
  e.g. [1-2] below -- do these works suffer from the same limitations as the
  ones mentioned in lines 30-38?
- Have the authors considered higher-dimensional analogues of their lower bound
  -- for example for optimization on the hyperbolic space $\mathbb{H}^d$, or
  Hadamard manifolds of negative curvature? This feels out-of-scope, but it may
  be interesting to know whether the same ideas have direct implications for
  general negatively-curved spaces.

## Minor Typos/etc

- line 149: missing a "plane"
- line 151: might want to consistently call this the "plane" since 'models' for
  the hyperbolic plane aren't discussed?
- Theorem 3: first "at" should be "and"?
- line 212: "expected logarithm of the number of options"
- Proof of theorem 3: the specific simplification here requires $r \geq 1$;
  maybe express this in the hypotheses
- eqn before line 235: loose parenthesis in numerator

## Referenced works

[1] https://arxiv.org/abs/2002.04144

[2] https://arxiv.org/abs/2008.02252


**Time Spent Reviewing:**

4

---

> ### Author Response · Authors · 2021-08-09
> **Thank you for the helpful review!**
>
> Thank you for your enthusiasm for our main result, as well as the many helpful comments.
>
> Indeed, Theorem 2 should have $d$ replaced by the squared distance divided by the squared diameter of the domain. We will correct this in the next version. Also, as you suggest, what we mean by an accelerated method is a guarantee analogous to that of Theorem 2, and we will spell this out explicitly. The important point is that the first term decays exponentially in $-k/2 \sqrt{\mu/L}$ and in our setting we have $L/\mu = \Theta(r)$ and thus an accelerated method would be able to reach a nearly optimal point with $O(\sqrt{r} \log r)$ iterations. However our main result shows that at least $\Omega(r/\log r)$ iterations are necessary.
>
> Thank you for pointing us to references [1] and [2]. In [1], the main convergence guarantees depend on a parameter $\zeta$. Theorem 4 gives simplified conditions under which the algorithm outperforms that of Zhang and Sra, but note that they need the sectional curvature to be bounded in terms of the diameter (which is not the case in our example). This happens, for example when the discrepancy of the manifold (a measure of how far it is from Euclidean) goes to zero. Essentially, our lower bound works because the volume of negatively curved spaces grows exponentially at large distance scales. And both [1] and other works that achieve local acceleration are operating in a regime where the volume does not grow that fast, either because we are already close to the optimum or because the curvature is approaching zero. Reference [2] is about acceleration for finding $\epsilon$-approximately critical points in non-convex functions, so what they mean by acceleration is rather different. It is about improving the dependence on $\epsilon$.
>
> In terms of generalizations, it seems possible that we might eventually be able to prove analogous results for any negatively curved manifold where the curvature is bounded from above, by say -1. However it is not clear if such nice functions, like the squared distance to a point on the boundary, will work.

---

> > ### Comment · Reviewer_seq7 · 2021-08-21
> > **clarification**
> >
> > Thank you for clarifying the "no robust acceleration" claim. I would like to follow up here, as I have some lingering confusion about your result and its implications given this interpretation.
> >
> > It seems a little strange to me to claim that the query complexity lower bound implies "acceleration" is impossible based on the part of the bound associated to the local linear rate for GD on a strongly-convex objective -- possibly it is just me, but my own immediate association when reading "first-order accelerated method" is the O(1/k^2) rate of Nesterov's method (I would appreciate the authors clarifying this issue for me if my perception is off here). Obviously, this issue is merely semantic as far as lower bounds are concerned, but it tends to make me feel like the significance of the lower bound is broader (and, in my interpretation, slightly oblique to, although that might be an issue on my end) than this -- I would interpret this based on your clarification as showing that, in the noisy oracle setting, we cannot achieve rates that have the same constants (i.e. non-k-dependent parts) as in the euclidean setting, and in this sense giving some evidence to a problem posed by Zhang and Sra in [37] (e.g. comment after Theorem 12 on page 11). Since this feels slightly different to me than just "no acceleration", it seems like it should bear some additional discussion in the paper.
> >
> > Could you please provide some additional clarification for me on this issue? I might be mistaken about something here. Thanks!

---

> > > ### Author Response · Authors · 2021-08-21
> > > **your questions**
> > >
> > > Thanks for the questions and we appreciate the effort to dig into what our results mean and where they fit in.
> > >
> > > First, I don't think the correct interpretation of our results is that we are proving acceleration is "impossible based on the part of the bound associated to the *local* linear rate for GD on a strongly-convex objective". This may just be a confusion over wording, but acceleration just means different things when you are in the smooth vs. smooth and strongly convex case. See e.g. these notes
> > >
> > > https://ee227c.github.io/notes/ee227c-notes.pdf
> > >
> > > On page 45 they give a table summarizing the different bounds. What you're referring to as the $1/t^2$ rate is what acceleration means in the smooth case. And in the smooth and strongly convex case, acceleration means getting a linear rate of convergence which depends on the squareroot of the condition number rather than the condition number.
> > >
> > > Second, many of the works studying Riemannian acceleration study the smooth and strongly convex case. E.g. these ones do:
> > >
> > > http://proceedings.mlr.press/v75/zhang18a.html
> > > https://arxiv.org/abs/2001.08876
> > > https://arxiv.org/abs/2012.03618
> > >
> > > As an aside, I don't think it's fair to call the notion of acceleration in the smooth and strong convex case just a matter of improving the constants. For example, it's the same improvement that's furnished by using Krylov subspace methods (and using Chebyshev polynomials) rather than the power method, and that's just in the special case of minimizing a quadratic function. See e.g.
> > >
> > > http://blog.mrtz.org/2013/09/07/the-zen-of-gradient-descent.html
> > >
> > > Third, and maybe perhaps most importantly, I think the way of phrasing it as "we cannot achieve rates that have the same constants (i.e. non-k-dependent parts) as in the euclidean setting" is misleading. There is not a well-defined notion of what the non-k-dependent part means. Are you allowed to have an additive term that depends exponentially on the diameter, and other parameters of the problem? If so, then yes our results do not preclude having just a $1/k^2$. But this means that such results only kick in when $k$ is extremely large (i.e. you are trying to solve it to extremely high accuracy), so that the $1/k^2$ term dominates. This is exactly where local acceleration comes in. What's really going on is that you're so close to the optimum (and it took you a lot of steps to get there) that you're in effect zooming in on the Riemannian manifold around the optimum, and it becomes locally Euclidean. So yes, in that case you can recover Nesterov acceleration, but then again it's in a setting where everything is very close to flat. In contrast, our bound is about what happens on a global scale, and in particular how many oracle calls you need in a negatively curved case to close enough that things become almost flat.
> > >
> > > I think this is very valuable discussion, and we will incorporate this into the next version of our paper. Thanks again for the questions, and happy to clarify further in case it is helpful.

---

> > > > ### Comment · Reviewer_seq7 · 2021-08-22
> > > > **thanks**
> > > >
> > > > Thank you for the additional (patient) clarification. I take your points, and agree that including this kind of discussion in the revision will go a long way to emphasizing the precise novelty/implications of the result to a broad audience (especially the discussion in the final paragraph of your previous comment seems valuable here; but of course the clarifications regarding equation (2) and etc. as well).
> > > >
> > > > I now feel my own understanding of the result "compiles", and so raise my rating.

---

### Decision · Program_Chairs · 2021-09-27

**Decision:**

Accept (Poster)

**Comment:**

While reviewers felt the intuition behind the paper is relatively straightforward, ultimately they agreed that the results could provide useful  for practical applications.